# Use of oral anticoagulants in older people with atrial fibrillation in UK general practice: protocol for a cohort study using the Clinical Practice Research Datalink (CPRD) database

Anneka Mitchell [iD] ,[1,2] Tomas J Welsh,[3,4,5] Margaret C Watson,[6] Julia Snowball,[1] Anita McGrogan[1]

For numbered affiliations see end of article.

**Correspondence to**
Anneka Mitchell;
a.mitchell@bath.ac.uk

## ABSTRACT

**Introduction** Warfarin has frequently been underused in older people for stroke prevention in atrial fibrillation (AF). Direct oral anticoagulants (DOACs) entered the UK market from 2008 and have been recommended as an alternative to warfarin. This study aimed to describe any changes in the prescribing of oral anticoagulants (OACs) to people aged ≥75 years in UK general practice before and after the introduction of DOACs, to examine differences in patient characteristics which may influence prescribers' decisions regarding anticoagulation, to evaluate the time people stay on OACs and switching between OACs.

**Methods and analysis** A retrospective cohort study design will be used. Patients with a diagnosis of AF will be identified from the Clinical Practice Research Datalink (CPRD). The study period will run from 1 January 2003 to 27 December 2017. Patients enter the cohort at the latest date of the start of the study period, first AF diagnosis, 75th birthday or a year from when they started to contribute research standard data. Follow-up continues until they leave the practice, death, the date the practice stops contributing research standard data or the end of the study period (27 December 2017). Exposure to OACs will be defined as ≥1 prescription issued for an OAC of interest during the study period. Patients issued an OAC in the year preceding study entry will be defined as 'prevalent users'. Patients starting on an OAC during the study period will be defined as 'incident users'. Incidence and prevalence of OAC prescribing, patient demographics and characteristics will be described during three time periods: 2003–2007, 2008–2012 and 2013–2017. Persistence (defined as the time from initiation to discontinuation of medication) with and switching between different OACs will be described.

**Ethics and dissemination** The protocol for this study was approved by the CPRD Independent Scientific Advisory Committee. The results will be disseminated in a peer-reviewed journal and at conferences.

**Trial registration number** EUPAS29923.

## INTRODUCTION

A large body of evidence has shown that vitamin K antagonists (VKAs) are safe and effective for stroke prevention in atrial

---

**Strengths and limitations of this study**

► This will be a large cohort study using a validated data source which is representative of the UK population.
► This study will explore a number of aspects of oral anticoagulant (OAC) prescribing and will examine changes over time to show any changes in OAC prescribing practices since the introduction of direct oral anticoagulants (DOACs).
► Analysing differences in patient demographics, comorbidities and coprescribing between those prescribed DOACs and warfarin will provide important information for future studies aiming to compare differences in outcomes with the different OACs.
► This study relies on prescription data generated by general practices; therefore, we will not know if the prescribed medication was ever dispensed or taken.
► Warfarin dosing schedules are not recorded in the Clinical Practice Research Datalink, so duration of warfarin prescriptions will be estimated.

---

fibrillation (AF), but they have frequently been underused in older people.[1] Reasons cited by physicians as making them less likely to prescribe VKAs for a person with AF include advancing age, risk of falls, comorbidities and previous bleeding.[2]

The first direct oral anticoagulant (DOAC), dabigatran, was marketed throughout Europe and in the UK in 2008, but was not licensed for stroke prevention in AF until 2011. Since then, three additional DOACs have been licensed for prevention of stroke: rivaroxaban, apixaban and edoxaban, and these are recommended in national and international guidelines as an alternative to warfarin.[3–5] In the UK, the National Institute of Clinical Excellence (NICE) produces technology appraisals making recommendations for new medicines. The National Health Service is

BMJ

legally obliged to fund those medicines recommended by NICE within 3 months of publication of the appraisal. NICE published favourable technology appraisals for dabigatran and rivaroxaban in 2012,[6 7] apixaban in 2013[8] and edoxaban in 2015,[9] meaning DOACs were available across the UK from mid-2012 onwards for stroke prevention in AF.

DOACs may allay some prescribers' concerns when deciding whether to anticoagulate their older patients: they have fixed dosing schedules and can be added to compliance aids. They have also been shown in clinical trials to have a significantly lower risk of intracranial haemorrhage than warfarin[10–13] (a prominent concern in patients with a history of falls).[14]

Since the introduction of DOACs, the overall rate of oral anticoagulant (OAC) initiation has increased by 58%.[15] However, it is not known whether the introduction of DOACs has changed the rates of OAC prescribing for older people (aged ≥75 years) or how patient demographics, comorbidities and concomitant prescribing affect the choice of anticoagulant prescribed in UK general practice.

This study aimed to characterise prescribing of OACs to people aged ≥75 years in UK general practice, before and after the introduction of DOACs.

## Objectives

1. To describe changes in the point prevalence and incidence of OAC prescribing by year prior to the introduction of DOACs (2003–2007), between the introduction of DOACs and the time they were recommended by NICE for stroke prevention in AF (2008–2012), and following NICE recommendation (2013–2017).
2. To describe switching between OACs during the study period.
3. To compare the characteristics of patients with AF who were newly started on OACs during each period described in objective 1 to those who were not prescribed anticoagulation in the same period.
4. To describe persistence with DOACs compared with warfarin.

## METHODS AND ANALYSIS

This will be a retrospective cohort study using routinely collected healthcare data from the Clinical Practice Research Datalink (CPRD) database. The CPRD contains anonymised medical records and prescribing data from general practitioners (GPs) in primary care. It contains data for around 7% of the UK population and is representative in age, sex and ethnicity.[16] Sociodemographic data, medical diagnoses, other clinical and test data are recorded using Read codes, and these Read codes will be used to identify patients to be included in the study and also to identify covariates as described further.

## Source population

The study period will start on 1 January 2003 and will end on 27 December 2017. The source population will consist of all patients in CPRD who have ≥1 Read codes for AF and have contributed a minimum of 12 months' research standard data prior to entry into the cohort. To be eligible to join the study, patients need to have the following: a Read code for AF recorded during or any time prior to the start of the study period; evidence to support the diagnosis of AF, such as a second AF diagnosis at a later date, an echocardiogram or prescription for a rate limiting or antiarrhythmic medication in the 3 months before the AF diagnosis or anytime after; and age of ≥75 years at study entry.

Patients will enter the cohort at the latest of the following dates:
► Start of the study period.
► First AF diagnosis.
► 75th birthday.
► A year from the date they started to contribute research standard data.

Patients will be censored if they leave the practice, the practice stops contributing data, at their date of death or at the end of the study. For the incidence calculations and comparison of patient characteristics, patients will be censored when they are prescribed an OAC.

## Time periods

The cohort will be analysed in three time periods:
► Period 1: prior to the introduction of DOACs (2003–2007).
► Period 2: during the period between the introduction of DOACs and the time they were recommended by NICE (2008–2012).
► Period 3: following the publication of the NICE technology appraisals recommending DOACs as an option for stroke prevention in AF (2013–2017).

For each time period, patients will contribute data to either the warfarin, DOAC or no OAC group. Patients in the cohort can contribute data to more than one time period and more than one group, and may contribute differently for each objective. Patients may only contribute to the incident OAC group once with their index OAC. Any further prescriptions for different OACs would be considered to be prevalent.

## Exposure

The OACs of interest in this study are warfarin, dabigatran, rivaroxaban, apixaban and edoxaban. Warfarin is the only VKA included as it is the most commonly used VKA in the UK. Dabigatran, rivaroxaban, apixaban and edoxaban are the only DOACs licensed for the prevention of stroke in AF available during the study period.

All prescriptions for the OACs of interest that are issued following an AF diagnosis, during the study period and in the year prior to entry in to the study cohort will be identified.

## Incident and prevalent users

Exposure status will be defined at cohort entry as

- ► 'No OAC', where the patient had at least a year free of OAC prior to study entry and does not receive an OAC during the study period.
- ► 'Incident OAC users', where the patient had at least a year free of OAC prior to study entry then was prescribed an OAC during the study period.
- ► 'Prevalent OAC users' if the patient received an OAC in the year prior to study entry.

Exposure status for incident OAC users will be stratified at the start of each study period to no OAC, warfarin or DOAC.

### Indication for prescribing

OACs are prescribed for many different indications, but only those prescribed for stroke prevention in AF will be included in this study. Patients will be excluded from the study if

- ► They had a hip fracture or hip replacement in the 6 weeks prior to the OAC index date and only one prescription for an OAC.
- ► They had a venous thromboembolism in the 6 months prior to the OAC index date.

### DOAC prescription duration

The index date will be defined as the date the first prescription was issued. The expected end date will be calculated by dividing the number of tablets prescribed by the licensed number of doses per day (ie, two doses per day for dabigatran and apixaban and one dose per day for rivaroxaban and edoxaban). Where the quantity of tablets prescribed is not specified, the expected duration will be imputed from other prescriptions issued for the same product for that patient or overall for that product if patient-specific information is not available.

Patients will be classified as being continuously exposed until there is a gap of >60 days between the expected end of one prescription and the start of the next. Where there is a gap of ≤60 days between the expected end of one prescription and the start of another for the same DOAC (regardless of whether the strength prescribed changes), the gap will be filled and defined as continuous exposure. Where a subsequent prescription for the same product is issued before the expected end of the previous prescription, the prescriptions will be concatenated and the extra days will be added to the expected end date of the second prescription. A maximum of 60 additional days will be allowed per patient.

A DOAC will be defined as discontinued 60 days from the expected end date of the last prescription issued for that product.

### Warfarin prescription duration

Warfarin does not have a fixed dosing schedule; the dose is amended based on the results of a blood test, the international normalised ratio (INR). In the UK, patients are issued with different strength tablets (0.5, 1.0, 3.0 and 5.0 mg) and are advised, following each INR test, what dose they need to take and what strength of tablets to use. Dosing information

for warfarin is not recorded in the CPRD; however, the strength and quantity of warfarin tablets issued by the general practice are recorded. Read codes for warfarin monitoring and results of INR tests are also recorded by some general practices, but this is not recorded consistently.

The index date will be defined as either the date of the first prescription issued or from the date of the first INR test result above 2 if this occurs before the first issue of a prescription. Patients may be exposed to warfarin before the first prescription issued by their GP if treatment is started in the hospital. CPRD data do not include hospital prescribing; hence, INR test results of ≥2 will be used as a surrogate marker of exposure.

The expected end date for warfarin prescriptions will be estimated using the median time between all previous prescriptions issued to the same patient. For example, if four prescriptions had been issued for 28×1 mg tablets and the gaps between the prescription issue dates were 15, 35 and 42 days, we would define the final prescription for 28 tablets as lasting 35 days, so the expected end date of the last prescription would be 35 days from the issue date. Where multiple strengths are issued for a patient, the median will be calculated for each strength and the expected end date will be the latest of these dates.

If only one prescription has been issued, the expected end date will be assumed to be equal to the median warfarin prescription gap for the cohort.

A patient will be assumed to have discontinued warfarin treatment if there is a period ≥2 times their median prescription gap. If an INR test result of ≥2 is recorded in the discontinuation period (expected end date of the last prescription to the date equal to twice their median prescription gap), then INR tests will be used as a surrogate marker for warfarin prescribing and the end date will be defined as 1 week following the final INR test result of ≥2.

The procedure for warfarin mapping is illustrated in figure 1. This will be repeated for each strength and the patient classed as unexposed from the latest discontinuation date of all the strengths.

### Restarting index OAC after discontinuation

There could be instances where OAC therapy is classed as discontinued when in fact the OAC has been continued but not issued by a GP if the patient is hospitalised. In instances where patients are classified as discontinuing OAC therapy but have then been 'restarted' on their index OAC, they will be reviewed individually to ascertain whether the OAC therapy had truly been discontinued or whether their records indicate another reason for the gap in prescribing, for example, a record indicating an admission to hospital.

### OAC switching

The first OAC prescribed during the study (for incident users) or on entry to the study (for prevalent users) will be defined as the index OAC. If ≥1 prescription for an alternative OAC is prescribed during the study period, the patient will be designated as having switched. Switches can be from

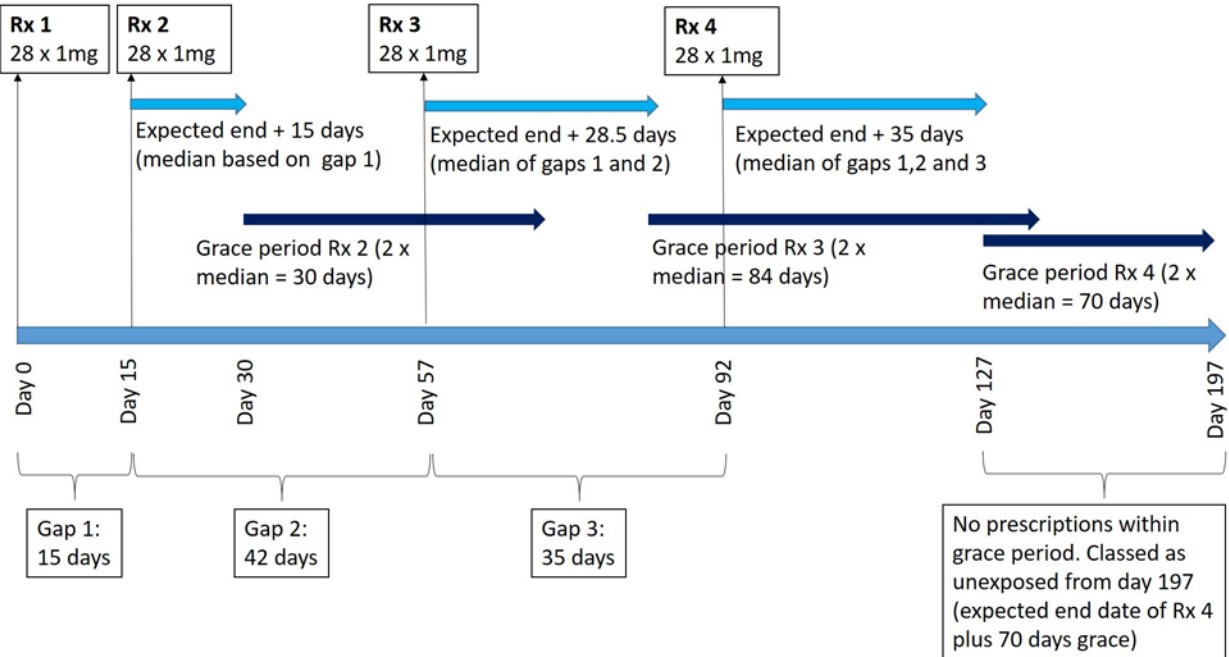

**Figure 1** Illustration of warfarin mapping for a single patient and one strength of warfarin tablet. Rx, prescription.

- ► Warfarin to DOAC.
- ► DOAC to warfarin.
- ► DOAC to alternative DOAC.

Patients will be assumed to have stopped the index OAC on the date the alternative OAC is issued if this is prior to the expected end date of the index OAC. Where the alternative OAC is prescribed after the expected end date of the index OAC, the time to switch will be described.

### Covariate classification

It is recognised that some patients may remain in the cohort for a long time and their covariates may change. Therefore, covariates will be identified separately for each of the time periods the patient contributes data and will be grouped by DOAC, warfarin or no OAC, depending on exposure for that period.

We want to investigate whether certain comorbidities influence the prescribing of OACs. We will therefore look for comorbidities accumulated up to the end of the time period for patients who do not receive an OAC or the OAC index date where one is prescribed. Patients will be classified as having a comorbidity if the patient record contains ≥1 diagnostic Read code for a disease or ≥3 Read codes supporting the presence of the disease at any time prior to the relevant date for that period. Read codes used to support the presence of disease include attendance at specialist clinics (eg, diabetes clinics) or codes related to follow-up, such as hypertension monitoring.

Concomitant medication will be classified as ≥1 prescription issues for a medication of interest in the 3 months prior to study entry, commencing an OAC, or the end of the time period. We will describe medications at study entry as baseline and then in each time period for the no OAC and incident OAC patients as described for the comorbidities previously mentioned. The medications of

interest affect either stroke or bleeding risk or commonly interact with OACs (antihypertensives, antiplatelets, statins, diabetic medications, antiarrhythmics, non-steroidal anti-inflammatory drugs, corticosteroids, anticonvulsants, systemic azoles, HIV protease inhibitors and ciclosporin).

### Definitions of covariates

Differences in patient demographics (age and biological sex) and characteristics (body mass index (BMI), smoking status and alcohol consumption) between those anticoagulated for AF and those not anticoagulated will be described for each period. Differences in the stroke risk score (Congestive heart failure, hypertension, age ≥ 75 (doubled), diabetes mellitus, stroke (doubled), vascular disease, age 65-74, sex category (female) ($CHA_2DS_2$-VASc))[17] and the bleeding risk score (Hypertension, abnormal renal or hepatic function, stroke, bleeding, labile INRs, elderly, drugs or alcohol (HAS-BLED)),[18] comorbidities and concomitant medication will also be described. These covariates are defined in the following subsections.

#### $CHA_2DS_2$-VASc score

International guidelines recommend using this score to calculate stroke risk in patients with AF.[3–5] Higher scores indicate an increasing risk of stroke and the maximum score is 9. Definitions for each element were based on those described by Lip and colleagues[17] but have been modified for clarity:

- ► *Congestive heart failure (1 point)*: right or left ventricular failure or heart failure. Supportive information includes Read codes for heart failure clinic attendance or referral, heart failure monitoring or reference to heart failure therapy/education.

► *Hypertension (1 point)*: primary or secondary hypertension or hypertensive disease states, for example, hypertensive heart disease. Supportive information includes codes for monitoring, referral to specialist clinics or reference to therapy for hypertension.

► *Age≥75 (2 points)*.

► *Diabetes mellitus (1 point)*: type 1 or type 2 diabetes mellitus diagnosis or the consequences of diabetes mellitus, for example, diabetic retinopathy. This also includes anyone prescribed oral hypoglycaemic agents and/or insulin, and excludes gestational diabetes and steroid-induced diabetes as these are often reversible. Supportive information includes dietary advice, reference to diabetes therapies, referral to specialist clinics or educational programme.

► *Stroke/transient ischaemic attack or thromboembolism (2 points)*: stroke includes ischaemic stroke transient ischaemic attacks and excludes haemorrhagic stroke and venous infarctions. Thromboembolism includes arterial embolic disease and surgeries to treat arterial emboli, and excludes thrombosis and venous embolic disease, which are included in vascular disease.

► *Vascular disease (1 point)*: includes coronary artery disease (prior myocardial infarction, angina pectoris, percutaneous coronary intervention or coronary bypass surgery and aortic plaque); peripheral vascular disease (intermittent claudication, previous surgery or percutaneous intervention on the abdominal aorta, bifurcation or the lower extremity vessels (limited to surgery on the femoral, popliteal, peroneal and iliac arteries)); and arterial and venous thrombosis and excludes thromboses related to pregnancy.

► *Age 64–74 (1 point)*.

► *Female sex (score 1)*.

## HAS-BLED

International guidelines recommend using this score to assess the risk of bleeding with anticoagulants.[3–5] Higher scores indicate an increased risk of bleeding with anticoagulants. The presence of each disease scores one point and the maximum score is 9. For the purpose of this study, the maximum score for the HAS-BLED will be 8. The L in HAS-BLED corresponds to labile INR defined as a time in therapeutic range with warfarin of less than 60%. As INRs are not recorded consistently in the CPRD and because people prescribed DOACs would not have INR tests at all, this element of the score will not be included. Definitions for each element were based on those described by Pisters and colleagues[18] but have been adapted to suit the information recorded in the CPRD.

► *Hypertension (1 point)*: as described for $CHA_2DS_2$-VASc score previously.

► *Abnormal renal or hepatic function (1 point each, maximum of 2 points)*: renal and hepatic function will be determined using a combination of clinical Read codes and test results.

– Renal impairment: includes chronic dialysis, transplant and chronic renal impairment (defined as stage 3 or above, moderate, severe or end stage). If renal impairment is recorded with no severity but there is a test result for creatinine of ≥200 μmol/L, the patient will be allocated a point. If there are no test results recorded, the patient will score 0. Codes for renal disease where it does not state that renal function is impaired will be excluded.

– Hepatic disease: includes chronic hepatic disease (cirrhosis, hepatitis, fatty liver and fibrosis), liver transplantation or biochemical evidence of significant hepatic derangement (bilirubin>2× upper limit of normal in association with aspartate aminotransferase/alanine aminotransferase/alkaline phosphatase) where these blood test results are recorded in CPRD.

► *Stroke (1 point)*: as described for $CHA_2DS_2$-VASc score previously.

► *Bleeding history or predisposition (1 point)*: includes any previous bleed; categorised based on the International Society of Thrombosis and Haemostasis guidance as major bleeding where the bleed occurs in a critical area or organ or requires a blood transfusion (CPRD does not state the number of units infused, so any requirement for a blood transfusion in conjunction with a Read code for bleeding will be classified as a major bleed)[19]; or clinically relevant non-major bleeding (defined as a bleed not fitting the major bleed criteria but requiring review by a healthcare professional, as the codes are recorded by a GP we will assume that these bleeds required review)[20]; excludes codes for traumatic bleeding, bleeding related to pregnancy and postoperative bleeding. Predisposition to bleeding includes anaemia and gastric, duodenal and peptic ulcers.

► *Elderly (1 point)*: defined as being aged >65 years old.

► *Drugs or alcohol (1 point each, maximum of 2 points)*: a point will be allocated where a patient receives a prescription for an antiplatelet or non-steroidal anti-inflammatory medication within the time period. Exact alcohol consumption is not well documented in the CPRD, but patients will score a point for this element if they have any record stating the number of units consumed per week as >8 units or if they have any codes for heavy/problem drinking.

## Renal impairment

DOACs are predominantly renally cleared and so require dose adjustment and careful monitoring in renal impairment. The elimination of warfarin is mostly metabolic, being metabolised by hepatic enzymes so it is not affected by changes in renal function. Renal impairment may therefore affect the decision of which anticoagulant (if any) to use so it has been included as a covariate alone, in addition to its inclusion in the HAS-BLED score. The Read code or test result dated closest to the relevant date for the period will be used to categorise the presence and level of renal impairment.

### Chronic renal impairment

Includes chronic dialysis, renal impairment and chronic kidney disease regardless of whether severity or stage is reported. Test results for creatinine and estimated glomerular filtration rate (eGFR) will also be obtained where available. If test results are available, they will be used to classify whether a patient has renal impairment and the stage of impairment (if eGFR is documented). Codes for a renal disease that may reduce kidney function but with no test or diagnostic code to support the presence of renal impairment will be excluded.

### Acute kidney injury

Previous episodes of acute kidney injury may influence the decision to prescribe an anticoagulant and which anticoagulant to prescribe. Read codes for acute kidney injury will be counted as discrete events where they occur ≥1 week apart.

### Hepatic disease

Hepatic disease is an independent risk factor for bleeding and also affects the metabolism of warfarin, so it may affect the decision to prescribe anticoagulation. Hepatic disease will be defined as specified under the HAS-BLED section.

### Previous bleed

Previous bleed includes any non-traumatic bleeding or haemorrhage. Bleeds will be divided in to subsections of major bleeding and clinically relevant non major bleeding (as described in HAS-BLED section), and also by site of bleed (intracranial bleeds, gastrointestinal bleeds or others).

### Dementia

Dementia include dementia (any type), Alzheimer's disease and anyone prescribed an acetylcholinesterase inhibitor or memantine and excludes cognitive impairment where there is no diagnosis of dementia.

### History of or at risk of falls

This includes any type of fall and any Read code which specifically indicates the practitioner is concerned that the patient is at risk of falling and excludes gait disturbances and other factors which may increase risk of falling but have not been defined by the GP as causing concern regarding falls risk.

### Fragility fractures

Fragility fractures are defined by the NICE as "…most commonly in the spine (vertebrae), hip (proximal femur) and wrist (distal radius). They may also occur in the arm (humerus), pelvis, ribs and other bones".[21] It is often not recorded in CPRD whether a fracture is thought to be a fragility fracture or not. Read codes for a diagnosis of fracture or operation to repair a fracture occurring in the spine, proximal femur, long bones, distal radius, humerus, pelvis or ribs will be included. Non-specific codes for osteoporotic and fragility fractures will also be included. Pathological fractures (unless specified that they are due to osteoporosis) will be excluded.

### Heart valve replacements and mitral stenosis

Historically, AF patients have been classed as having valvular or non-valvular AF, but definitions of these have varied.[22] The licensing for DOACs in the UK states they should only be prescribed for patients with non-valvular AF, however, the randomised controlled trials for the DOACs all defined valvular AF differently and exclusion criteria varied.[23] Recently updated European Society of Cardiology guidelines suggest that VKAs should be the preferred option for patients with mechanical heart valves or moderate to severe mitral stenosis.[4] This co-variate has been included to investigate what proportion on DOAC and warfarin users have a history of mechanical heart valves or Read codes for moderate or severe mitral stenosis.

### Number of general practice encounters

The number of encounters with healthcare professionals from general practice in the year prior to study entry, the year preceding the OAC index date or preceding the end of the time period will be used as an indicator of healthcare use.

### Statistical analysis

Objective 1: to describe changes in the point prevalence and incidence of OAC prescribing by year prior to the introduction of DOACs (2003–2007), between the introduction of DOACs and the time they were recommended by NICE for stroke prevention in AF (2008–2012), and following NICE recommendation (2013–2017).

Only patients prescribed an OAC will contribute data to the incidence and prevalence calculations. For incidence, patients may only contribute to either the warfarin or the DOAC group once as they will be censored at the time they are prescribed an OAC. The point prevalence will be calculated at the midpoint of each year so will take account of switching. For prevalence, a patient could contribute to the warfarin group for some years, and the DOAC group for other years. Prescription mapping will be used to calculate the number of people with an OAC prescription that includes that time point as the numerator and the total number of people in the cohort at that time point as the denominator.

The incidence of OAC prescribing will be calculated overall and for each specific OAC. Everyone in the cohort will be considered 'at risk' until the time they receive their first OAC prescription. Incidence will be calculated using the number of incident OAC users as the numerator and person-time at risk for non-exposed persons as the denominator. For the person-time at risk calculation, the denominator will be truncated at the date of their first OAC prescription. Incidence will be stratified by age (75–84 and 85+) and sex.

Figure 2 illustrates how patients may contribute to the incidence and prevalence calculations throughout the study.

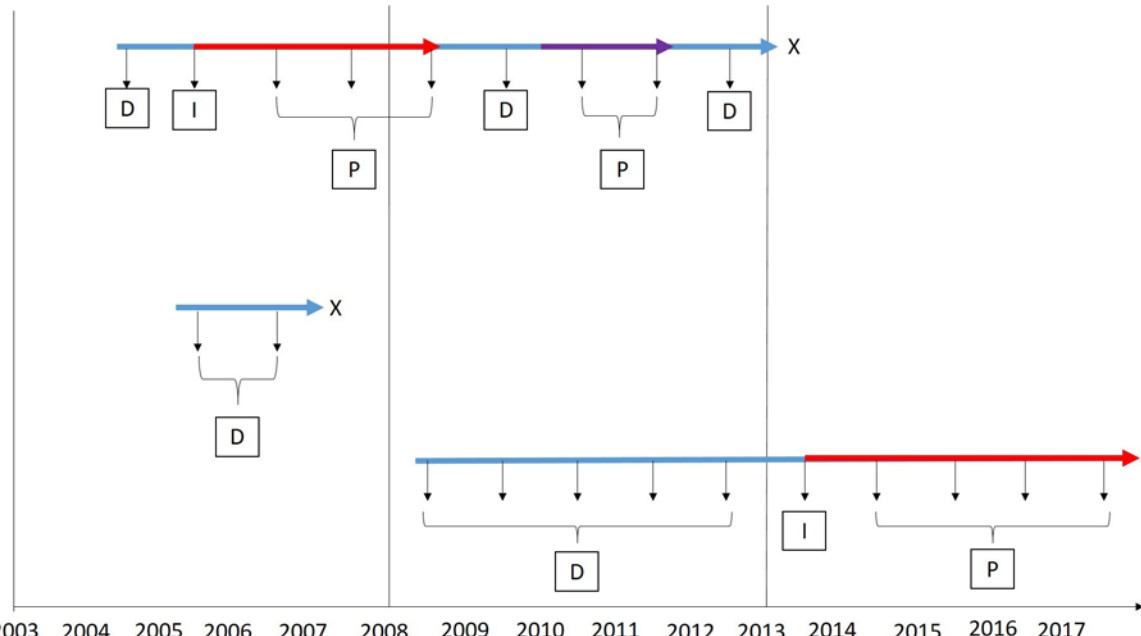

**Figure 2** Illustration showing how patients may contribute to the different groups for incidence and prevalence. The blue arrows show total unexposed time in the study; the red arrows indicate exposure to warfarin; the purple arrow indicates exposure to DOAC. The vertical arrows show the yearly incidence and prevalence calculations, and the letters in boxes correspond to the group the patient would contribute to for that year. Incidence and prevalence for the first D then prevalence only for all subsequent Ds. D, denominator; DOAC, direct oral anticoagulant; I, incidence numerator, P, prevalence numerator; X, left the study.

### Objective 2: to describe switching between OACs during the study period

The number and percentage of patients switching from warfarin to DOAC, DOAC to DOAC and DOAC to warfarin will be calculated. The number of patients with multiple switches and details of the switches will be described. The average time to switch will be reported.

### Objective 3: to compare the characteristics of patients with AF who were newly started on OACs during each period described in objective 1, to those who were not prescribed anticoagulation in the same period

Demographics, characteristics, comorbidities and concomitant medications will be reported for the three time periods described. Prevalent OAC users will be excluded from this part of the study as we want to compare those started on an OAC with those who are not during each time period.

Patients may contribute data to more than one group, but they will be censored at the time they are prescribed their index OAC. For example, they may enter the study during the first time period but may not be prescribed an OAC until period 3. In this case, they would contribute data to the no-OAC group in time periods 1 and 2 and then one of the OAC groups in period 3. If they entered in period 1 and started on an OAC in period 2, they would contribute data to the no OAC group in period 1 and the OAC group in period 2; they would not contribute any data to period 3 as they would be censored. Figure 3 illustrates how patients contribute to different groups during their time in the study. Patient

demographics (age and biological sex) and characteristics (BMI, smoking status and alcohol consumption) will be reported as percentages in predefined categories. The proportion of the cohort with each of the comorbidities and concomitant medications will be described. The mean $CHA_2DS_2$-VASc and HAS-BLED scores will be reported.

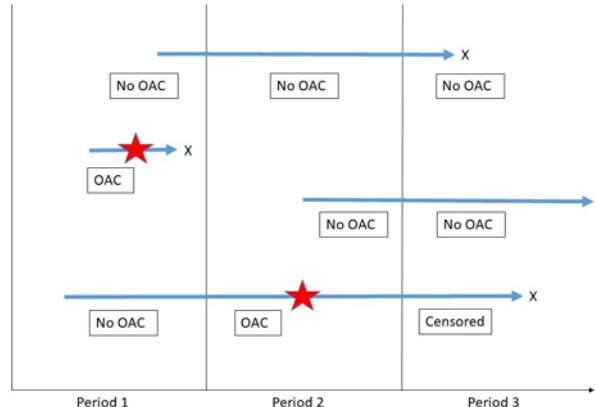

**Figure 3** Illustration showing how patients contribute to different groups for the characteristic comparisons during their time in the study. The blue arrows represent individual patients; the stars represent the date the index OAC was commenced; X represents the patient leaving the study. The boxes describe which group the patient will contribute comorbidity data to for that period (OAC will be subdivided to warfarin or direct oral anticoagulant, depending on which was prescribed). OAC, oral anticoagulant.

### Objective 4: to describe persistence with DOACs compared with warfarin

Duration of prescribing of DOACs will be compared with warfarin using Cox proportional hazards, stratified by DOAC and adjusted for age and other important covariates identified in objective 3. All incident OAC users will be included. Time on OAC will be calculated as the difference between the discontinuation date of the OAC and the index date, as defined previously. Patients will be censored at the outcome (discontinuation of OAC therapy), on the date they stop contributing research standard data or the date of death.

All analyses will be conducted in STATA V.15.[24] All values will be reported with 95% CIs.

### Missing data

There is likely to be a substantial amount of missing data for weight, BMI, smoking status and alcohol consumption as these variables may not be regularly updated by GP practices. We will use logistic regression to ascertain whether other variables (age, sex or comorbidities) predict whether these data are missing. If these variables do not predict the missing data, then we will assume that the data are missing at random. The proportion of missing data for these variables will be presented. If the data are missing at random, we will impute the missing values using multiple imputation and conduct a sensitivity analysis to investigate the effect of the missing data on the results of the analyses. However, previous studies suggest it may only be appropriate to impute weight and BMI.[25 26]

### PATIENT AND PUBLIC INVOLVEMENT

Patients and the public were not involved in the production of this protocol or in the design of the study.

### DISCUSSION

A number of studies have described changes in anticoagulation prescribing in the time since DOACs were introduced in the UK.[15 27 28] However, none have focused specifically on their use in older people aged ≥75 years, despite older people being the largest users of anticoagulants for AF. The main strength of this study is that it will investigate prescribing in a large cohort of older patients using the CPRD, which is generally representative of the UK population. This study will look at a number of different factors that may influence the prescribing of anticoagulants and how these have changed over time. The results of this study will inform future comparative studies between OACs as it will provide an insight to potential confounders which may need to be controlled for. A limitation of this study is defining accurately when patients are exposed to an anticoagulant and when they are not. This is particularly difficult for warfarin as dosing information is not included in the CPRD. A number of studies have compared DOACs and warfarin using routinely collected healthcare data, but they do not describe how they mapped warfarin exposure.[29 30] This protocol is transparent in describing our planned method for warfarin mapping and uses both prescription and test result data. We will also investigate gaps in prescribing to establish whether patients may have been in the hospital or having their OAC prescribed elsewhere; this is likely to be more common in this age group than others and has been noted as a limitation in other studies.[31] A second limitation is that comorbidities may not always be recorded in the CPRD, particularly where they are diagnosed and treated in secondary care. However, as the comorbidities we have chosen are chronic and likely to have a significant impact on patients' health, it is likely that they would be recorded accurately. Finally, as it is difficult to predict whether prescriptions for concomitant medicines are still continuing at any given time, we have decided to assume that if a prescription was issued in the 3 months prior to an OAC being started, the concomitant medicine is still continuing at that time. This is likely to be the case as most medications of interest are for chronic conditions. However, we acknowledge that it may overestimate exposure to these concomitant medications.

### ETHICS AND DISSEMINATION

The protocol for this study was approved by the CPRD Independent Scientific Advisory Committee (protocol 18_071) and is registered with the European Network of Centres for Pharmacoepidemiology and Pharmacovigilance. Results of the study will be disseminated in a peer-review journal with open access available within 6 months of publication. Results will also be communicated at national and international conferences as appropriate.

**Author affiliations**
[1]Pharmacy and Pharmacology, University of Bath, Bath, UK
[2]Pharmacy Research Centre, University Hospital Southampton NHS Foundation Trust, Southampton, Southampton, UK
[3]Research Institute for the Care of Older People, Bath, UK
[4]Institute of Clinical Neurosciences, University of Bristol, Bristol, UK
[5]Older Persons Unit, Royal United Hospitals Bath NHS Foundation Trust, Bath, Somerset, UK
[6]Strathclyde Institute of Pharmacy and Biomedical Sciences, University of Strathclyde, Glasgow, UK

**Contributors** AM, TJW, MCW, JS and AMcG all contributed to the design of the study. AM drafted the protocol, which was then reviewed and revised by TJW, JS, MCW and AMcG.

**Funding** This work was supported by The Dunhill Medical Trust (grant number RTF109/0117).

**Competing interests** None declared.

**Patient consent for publication** Not required.

**Provenance and peer review** Not commissioned; externally peer reviewed.

**ORCID iD**
Anneka Mitchell http://orcid.org/0000-0002-9222-3185

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
