## [Reviewer comments · BMJ Open]

ARTICLE DETAILS

TITLE (PROVISIONAL)	Utilisation of oral anticoagulants in older people with atrial fibrillation in UK general practice: a protocol for a cohort study using the Clinical Practice Research Datalink (CPRD) database
AUTHORS	Mitchell, Anneka; Welsh, Tomas; Watson, Margaret; Snowball, Julia; McGrogan, Anita

VERSION 1 – REVIEW

REVIEWER	Madeleine Durand Université de Montréal, Québec, Canada
REVIEW RETURNED	18-Sep-2019

GENERAL COMMENTS	The authors describe the protocol of a retrospective cohort study conducted in the CPRD. The study will be a description of the use of oral anticoagulants in atrial fibrillation in the elderly (>75), and its variation across three key time-periods in the UK : before the licensing of DOACs, after DOAC licensing but before NICE recommendation of DOACs for anticoagulation in atrial fibrillation, and following NICE recommendation. Characteristics of patients prescribed DOACS vs. Warfarin, persistence on drugs and patterns of switch will also be studied. This is an important study and it will yield important clinical information. Please find below some comments on this protocol, which I believes requires some clarifications before it is ready for publication: 1. In the methods section, authors mention that AF will be defined using the presence of a Read code for AF, AND “evidence to support the diagnosis of AF such as a second AF diagnosis at a later date, an echocardiogram or prescription for a rate limiting or antiarrhythmic medication in the 3 months before the AF diagnosis or any time after”. This two-step definition, with study entry being defined at the time of the initial AF Read code, will introduce some immortal time. In view of the objectives of the study, I do not think this immortal time will introduce much bias. However, authors might re-consider this two-step definition. If they want to keep it, it would be more prudent to start follow-up when the definition for Afib is met (i.e.: on the date of the second criteria when it happens after the initial AF read code.)2. Please place the section defining the criteria for end of follow-up directly after the list of criteria for start of follow-up; it will make the text easier to follow.3. In the exposure section, it is stated that : “For each time period, patients will contribute data to either the warfarin, DOAC or no
--

	OAC group depending on whether they were prescribed an OAC during that period.” It must be stated how patients will be classified when they switch or stop during the 3 study periods? Will exposure status be defined at the start of the study period exclusively, or for incident patients, at cohort entry date? Please specify. The 3 study periods span 3-4 years each, so there will be lots of switching and stopping within each. Exposure rules must be very clear, and a figure could be very helpful to help understand the different case scenarios. E.g. What would be the exposure status of this patient for the second time period : He enters the study on March 2008 with a diagnosis of AF and cardiac ultrasound. On January 2011 he gets a prescription for warfarin, which he takes for 6 months than stops, and on June 2011 he starts dabigatran which he continues thereafter. For most of period 1, he was unexposed to OAC, then he got a short exposure to warfarin and switched to a DOAC. ** This is my main concern with this protocol** 4.- In the Indication for prescribing section, it is stated that “Patient records will be investigated further” if other reasons (post surgical period or venous thromboembolism) are identified for OAC use. This is imprecise. What does investigated further mean, and how will these cases be handled in the analysis? Will the prescriptions that meet those criteria be removed? If so, a patient with AF who undergoes a knee surgery would be considered to stop his/her OAC for stroke prevention following surgery? This could affect persistence data... If a rule is designed to remove some prescriptions, it must be clearly stated here. I must say that as patients enter the cohort with a diagnosis for AF, my personal preference would be to consider all later prescriptions of OACs as related to AF, even if that will lead to some misclassification. Given CPRD’s size, this study will include huge numbers : looking at individual charts to individualize decisions seems both imprudent (hard to standardize) and herculean. 5. In the DOAC prescription duration section. One paragraph mentions that a grace period of 30 days following the estimated end date of a prescription will be granted, then the other paragraph says that “A DOAC will be defined as discontinued 60 days from the expected end date of the last prescription issued for that product”. Those two are contradictory. Both durations are reasonable but they must be consistent. What is not considered continuous use should be considered discontinuation. 6. When defining warfarin prescription duration, the word “index date” is used. It would be better to use either start date or index date throughout, but the switch between the two terms is confusing. As INR tests are used in defining warfarin start date, could authors say for how many of the participants they expect some recording of INR results? (They stated that this recording was inconsistent across practices.) 7. The definition of covariates section is hard to follow. It seems strange that covariates would be defined differently (timing-wise) for patients exposed or unexposed to OACs. Unexposed patients have an « arbitrary » date (by arbitrary I mean a date that is not meaningful to their personal medical history) at the beginning of each of the 3 study periods. Exposed patients’ covariates, on the other hand, are defined when they are prescribed an OAC. The
--	---

	fact that both diagnoses and prescriptions in CPRD are generated during doctors' visits increases the likelihood that those prescribed OACs (or any other drug actually) will also have more comorbidities registered on that day. As comparing covariates' profiles between the users and non-users of OACs is one of the main objectives, the timing of covariates definition is very important. I am also concerned that some covariates seem to be defined using the future, I think that is imprudent and should not be done. I would strongly favour a definition of covariates at the beginning of each of the 3 study periods for everyone regardless of their exposure status, and to define covariates based on diagnoses and treatments accumulated up to that date. ** This is my second most important concern ** 8. There could be a discussion section about strengths and weaknesses of the proposes study. Minor points Abstract : In methods and analysis :  - the words retrospective cohort study should be in there -the list that starts with « Follow-up continues until » should include the study end date; 27th of December 2017. - The following sentence is unclear: "Patients entering the study on an OAC and those issued an OAC in the year preceding study entry will be defined as 'prevalent users'." What is the difference between the two statements? Do you mean that those currently taking OACs at cohort entry date, or prescribed an OAC in the year preceeding cohort entry (even if they had stopped on the date of cohort entry) will be defined as "prevalent users"? In this case, it is sufficient to say: "All patients prescribed and OAC in the year prior to cohort entry date will be defined as 'prevalent users'."
--	---

REVIEWER	Angel Wong London School of Hygiene and Tropical Medicine, United Kingdom
REVIEW RETURNED	23-Oct-2019

GENERAL COMMENTS	This is a cohort study investigating the utilisation of OACs over time, describing the baseline characteristics of the OAC users in different time period, comparing OAC users and non-OACs users and describing characteristics of switchers among patients with AF using CPRD data. The protocol of this descriptive study is written very clearly. I only had some minor comments for authors to consider:  1. Can the study period extend to 31st December 2017 instead of 27th December 2017? 2. Authors will identify patients with AF before study period as well – will they also investigate the difference between incident AF/prevalent AF. Because people aged ≥ 75 years will only be included, I wondered if those had AF long time ago, their knowledge/perception of prescribing choice were different from incident AF that might be worth exploring? 3. Authors stated that they will identify Rx of OACs in the year prior to entry to the study cohort as prevalent OACs. How about those who had ever prescribed OACs 1 year before the follow-up started? I reckon OACs are lifelong prescriptions and unlikely to be stopped over 1 year; there will be only a few patients. However it
--

	might be still good to exclude them in the analysis/group them in another category? 4. I might have missed – but do OACs users have to be prescribed after AF diagnosis in order to be included? 5. Authors mentioned they will also explore whether there are other indications for prescribing – but how to deal with these in their analysis? 6. In the session of warfarin prescription duration: For example, if four prescriptions had been issued for 28 x 1mg tablets and the gaps between the prescription issue dates were 15, 35, and 42 days we would define 28 tablets as lasting 35 days so the expected end date of the last prescription would be 35 days from the issue date. For example, if four prescriptions had been issued for 28 x 1mg tablets and the gaps between the prescription issue dates were 15, 35, and 42 days we would define 28 tablets as lasting 35 days so the expected end date of the last prescription would be 35 days from the issue date. I'm not sure why the 28 tablets Rx would be defined as 35 days based on the gaps between Rx dates (15, 35, 42 days). A figure to illustrate this will be very helpful. 7. Authors stated that they will investigate whether there was a record indicating an admission to hospital during the period of discontinuation – wondered how reliable the record, the best way is to use HES data? 8. P.7 Can the authors suggest examples of codes indicating supporting the presence of the disease? 9. Concomitant medications – do they have to have overlapped duration with OACs? Which medications will be investigated? 10. P.8 Hypertension – apart from Read codes, how about blood pressure records? 11. P.12 Missing data will be assessed for randomness using a validated method – can the authors provide a citation?
--	--

VERSION 1 – AUTHOR RESPONSE

Reviewer 1

Reviewer Name: Madeleine Durand

Institution and Country: Université de Montréal, Québec, Canada

Please state any competing interests or state 'None declared': None to declare

Please leave your comments for the authors below

The authors describe the protocol of a retrospective cohort study conducted in the CPRD. The study will be a description of the use of oral anticoagulants in atrial fibrillation in the elderly (>75), and its variation across three key time-periods in the UK: before the licensing of DOACs, after DOAC licensing but before NICE recommendation of DOACs for anticoagulation in atrial fibrillation, and following NICE recommendation. Characteristics of patients prescribed DOACS vs. Warfarin, persistence on drugs and patterns of switch will also be studied. This is an important study and it will yield important clinical information.

Please find below some comments on this protocol, which I believes requires some clarifications before it is ready for publication:

Comment	Response	Revision location (marked copy)
1. In the methods section, authors mention that AF will be defined using the presence of a Read code for AF, AND “evidence to support the diagnosis of AF such as a second AF diagnosis at a later date, an echocardiogram or prescription for a rate limiting or antiarrhythmic medication in the 3 months before the AF diagnosis or any time after”. This two-step definition, with study entry being defined at the time of the initial AF Read code, will introduce some immortal time. In view of the objectives of the study, I do not think this immortal time will introduce much bias. However, authors might re-consider this two-step definition. If they want to keep it, it would be more prudent to start follow-up when the definition for Afib is met (i.e.: on the date of the second criteria when it happens after the initial AF read code.)	The two step criteria was implemented to ensure that included patients had AF and that the diagnosis was not just a spurious code. Therefore, we feel that it is important to retain the two step process to ensure validity of our cohort. As the supporting evidence may be recorded before the AF diagnosis (in the case of ECG's for example) we are reluctant to change the date to be that of supporting evidence. We agree that immortal time bias is an important consideration but as you say, it is unlikely to introduce much bias in the context of this study and it should not be differential between the groups. We thank you for raising this concern and will consider it in future comparative studies we have planned.	N/A
2. Please place the section defining the criteria for end of follow-up directly after the list of criteria for start of follow-up; it will make the text easier to follow.	Agreed.	Page 5: Moved under criteria for start of follow up as suggested.
3. In the exposure section, it is stated that : “For each time period, patients will contribute data to either the warfarin, DOAC or no OAC group depending on whether they were prescribed an OAC during that period.” It must be stated how patients will be classified when they switch or stop during the 3 study periods? Will exposure status be defined at the start of the study period exclusively, or for	Thank you for highlighting that this section needs further clarification. Patients will be censored at the time they are prescribed an OAC. In the example stated here he would contribute to the warfarin group for period 2 as he entered in period 2 and was prescribed an OAC in that period. He would then be censored so the dabigatran prescription would not be	Page 5: More detail has been added to the “Time periods” section. Page 6: Further detail has been added to the “Incident and prevalent users” section to explain how exposure will be defined both at cohort entry and at different time points. Pages 12 and 13: “Statistical analysis” section updated to clarify how patients can

incident patients, at cohort entry date? Please specify. The 3 study periods span 3-4 years each, so there will be lots of switching and stopping within each. Exposure rules must be very clear, and a figure could be very helpful to help understand the different case scenarios. E.g. What would be the exposure status of this patient for the second time period: He enters the study on March 2008 with a diagnosis of AF and cardiac ultrasound. On January 2011 he gets a prescription for warfarin, which he takes for 6 months than stops, and on June 2011 he starts dabigatran which he continues thereafter. For most of period 1, he was unexposed to OAC, then he got a short exposure to warfarin and switched to a DOAC. ** This is my main concern with this protocol**	recognised for the purposes of comparing characteristics of those started on an OAC. The rationale for this is that we want to look at the characteristics of patients at the time they start their first OAC compared to those who do not start an OAC so we are only interested in the first OAC they are prescribed for this objective. We will also look at switching and for this purpose we will map the exposure as the time on warfarin, the gap and then the time on dabigatran.	contribute data to different groups at different time points for each element of the study. Figures 2 and 3 illustrate further how patients can contribute data over the course of the study. Further detail has been added to the “Incident and prevalent users” section on page 5 to explain how exposure will be defined both at cohort entry and at different time points.
4. In the Indication for prescribing section, it is stated that “Patient records will be investigated further” if other reasons (post-surgical period or venous thromboembolism) are identified for OAC use. This is imprecise. What does investigated further mean, and how will these cases be handled in the analysis? Will the prescriptions that meet those criteria be removed? If so, a patient with AF who undergoes a knee surgery would be considered to stop his/her OAC for stroke prevention following surgery? This could affect persistence data... If a rule is designed to remove some prescriptions, it must be clearly stated here. I must say that as patients	The rationale for investigating these patients was to ensure that the first OAC prescription they received was for stroke prevention in AF and not for one of the other indications so as not to distort incidence figures. We intended to look more closely at these patients as we did not want to empirically exclude them when their OAC may have been for AF despite their other diagnoses. We do not intend to look again for VTE or surgery when prescription mapping so prescriptions would not be removed. Once we have established that the initial indication for the OAC was AF	Page 6: “Indication for prescribing” amended to be more specific and clearly state who will be excluded.

enter the cohort with a diagnosis for AF, my personal preference would be to consider all later prescriptions of OACs as related to AF, even if that will lead to some misclassification. Given CPRD's size, this study will include huge numbers: looking at individual charts to individualize decisions seems both imprudent (hard to standardize) and herculean.	we assume that remains the case. However, as you correctly state there was a reasonable number of patients so we looked at a sample and made the decision to exclude all patients who had a VTE in the 6 months prior to starting their OAC. We also decided to exclude patients with hip replacements where they had only one OAC prescription. This was decided as we could not be sure that the OAC had been started for AF. We decided to include knee replacement patients as their OAC for VTE prevention tends to be prescribed by the hospital so should not be in the CPRD data.	
5. In the DOAC prescription duration section. One paragraph mentions that a grace period of 30 days following the estimated end date of a prescription will be granted, then the other paragraph says that "A DOAC will be defined as discontinued 60 days from the expected end date of the last prescription issued for that product". Those two are contradictory. Both durations are reasonable but they must be consistent. What is not considered continuous use should be considered discontinuation.	Thank you for highlighting this discrepancy	Page 6: Amended to 60 days throughout the section.
6. When defining warfarin prescription duration, the word "index date" is used. It would be better to use either start date or index date throughout, but the switch between the two terms is confusing. As INR tests are used in defining warfarin start date, could authors say for how many of	Start date has been changed to index date throughout. The feasibility of using INR test results for warfarin mapping was raised when we applied to the Independent Scientific Advisory Centre to use the	Pages 7 and 13: Change from start to index date made.

the participants they expect some recording of INR results? (They stated that this recording was inconsistent across practices.)	CPRD data. We did some preliminary counts at that time and found that the majority of warfarin patients have at least one INR result and in our cohort around 85% of incident warfarin patients have ≥ 1 INR recorded.	
7. The definition of covariates section is hard to follow. It seems strange that covariates would be defined differently (timing-wise) for patients exposed or unexposed to OACs. Unexposed patients have an « arbitrary » date (by arbitrary I mean a date that is not meaningful to their personal medical history) at the beginning of each of the 3 study periods. Exposed patients' covariates, on the other hand, are defined when they are prescribed an OAC. The fact that both diagnoses and prescriptions in CPRD are generated during doctors' visits increases the likelihood that those prescribed OACs (or any other drug actually) will also have more comorbidities registered on that day. As comparing covariates' profiles between the users and non-users of OACs is one of the main objectives, the timing of covariates definition is very important. I am also concerned that some covariates seem to be defined using the future, I think that is imprudent and should not be done. I would strongly favour a definition of covariates at the beginning of each of the 3 study periods for everyone regardless of their exposure status, and to define covariates based on diagnoses and treatments accumulated up to	Thank you for highlighting that this section is difficult to follow. The rationale for having different timings for looking back at covariates was that something may have changed mid period that led to the patient being prescribed an OAC so we wanted to look back from the point the OAC was prescribed. As the unexposed patients would not have this date we arbitrarily chose to look back from the start of the study period for this group. We had not considered that as diagnoses are recorded during doctors' visits that OAC patients may have more comorbidities registered so we thank you for making this important observation. We still feel that it is important to record comorbidities recorded at the time of the OAC prescription so propose that we look at comorbidities and treatments accumulated up to the OAC index date. However, for the unexposed patients we will look back from the end of the study period as they may have accumulated comorbidities which could have deterred prescribers from starting an OAC during this time period.	Page 8: "Covariate classification" section updated to state that covariates for the no OAC group will be measured from the end of the period rather than the start. The rationale for using different dates for each of the groups have also been added. Only the concomitant medication could be defined using prescriptions up to 3 months in the future. We have amended this to only look back 3 months and not forwards.

that date. ** This is my second most important concern **		
8. There could be a discussion section about strengths and weaknesses of the proposed study.	There is a brief strengths and weaknesses section underneath the abstract but as suggested we have added a discussion section. We will report strengths and weaknesses fully in the results paper for this study.	Page 14: A brief discussion section has been added.
Abstract In methods and analysis : - the words retrospective cohort study should be in there		Page 2: Added to methods section of abstract.
-the list that starts with « Follow-up continues until » should include the study end date; 27th of December 2017.		Page 2: Added to methods section of abstract.
- The following sentence is unclear: "Patients entering the study on an OAC and those issued an OAC in the year preceding study entry will be defined as 'prevalent users'." What is the difference between the two statements? Do you mean that those currently taking OACs at cohort entry date, or prescribed an OAC in the year preceding cohort entry (even if they had stopped on the date of cohort entry) will be defined as "prevalent users"? In this case, it is sufficient to say: "All patients prescribed and OAC in the year prior to cohort entry date will be defined as 'prevalent users'."		Page 2: Amended as suggested.

Reviewer: 2

Reviewer Name: Angel Wong

Institution and Country: London School of Hygiene and Tropical Medicine, United Kingdom

Please state any competing interests or state 'None declared': None declared

Please leave your comments for the authors below

This is a cohort study investigating the utilisation of OACs over time, describing the baseline characteristics of the OAC users in different time period, comparing OAC users and non-OACs users and describing characteristics of switchers among patients with AF using CPRD data. The protocol of this descriptive study is written very clearly. I only had some minor comments for authors to consider:

1. Can the study period extend to 31st December 2017 instead of 27th December 2017?	Unfortunately, we only have data until 27th Dec 2017	N/A
2. Authors will identify patients with AF before study period as well – will they also investigate the difference between incident AF/prevalent AF. Because people aged ≥ 75 years will only be included, I wondered if those had AF long time ago, their knowledge/perception of prescribing choice were different from incident AF that might be worth exploring?	This is an interesting question, we agree that prescribing choices may differ for patients who have had AF for a long time compared to those with new AF. As we are looking solely at older patients they are likely to have moved around GP surgeries over their life course and so we could not be sure whether AF diagnoses recorded in CPRD were incident or prevalent. So, whilst we agree that this question warrants investigation, it may be better answered using a data source with connected follow up over the whole of adulthood rather than CPRD and is outside the scope of our study.	N/A
3. Authors stated that they will identify Rx of OACs in the year prior to entry to the study cohort as prevalent OACs. How about those who had ever prescribed OACs 1 year before the follow-up started? I reckon OACs are lifelong prescriptions and unlikely to be stopped over 1 year; there will be only a few patients. However it might be still good to exclude them in the analysis/group them in another category?	We recognise that patients may have received an OAC more than a year before study entry and hence may not be completely OAC-naïve at study entry. However, we think a year free of OAC is a reasonable time frame to then class them as incident users. We can be relatively certain that patients have been OAC free for a year as they need a year of up to standard data to be included in the study. We would be less certain as to whether patients were ever users of OACs as given their age they may have moved between different GP surgeries	N/A

	and hence prescribing information may not be complete. Our predominant interest is to investigate characteristics of patients at the time an OAC is commenced. Whilst it may be preferable for us to include only patients who are completely OAC-naïve and compare them to never OAC users it is likely to limit our sample size unnecessarily in this instance. However, for future comparative studies we would agree with your assertion and would aim to class ever-users separately to never-users.	
4. I might have missed – but do OACs users have to be prescribed after AF diagnosis in order to be included?	Yes, but this was not clear from the protocol	Page 5: Added under heading “Exposure”.
5. Authors mentioned they will also explore whether there are other indications for prescribing – but how to deal with these in their analysis?	Thank you for highlighting. More detail has been added regarding how these patients will be dealt with.	Page 6: “Indication for prescribing” amended to be more specific and clearly state who will be excluded.
6. In the session of warfarin prescription duration: For example, if four prescriptions had been issued for 28 x 1mg tablets and the gaps between the prescription issue dates were 15, 35, and 42 days we would define 28 tablets as lasting 35 days so the expected end date of the last prescription would be 35 days from the issue date. For example, if four prescriptions had been issued for 28 x 1mg tablets and the gaps between the prescription issue dates were 15, 35, and 42 days we would define 28 tablets as lasting 35 days so the expected end date of the last prescription would be 35 days	Thank you for highlighting that this section is not very easy to follow. Warfarin mapping is complicated and difficult to describe but we have added a figure as you suggest which we hope makes the method we have used clearer.	Page 7: Figure 1 added.

from the issue date. I'm not sure why the 28 tablets Rx would be defined as 35 days based on the gaps between Rx dates (15, 35, 42 days). A figure to illustrate this will be very helpful.		
7. Authors stated that they will investigate whether there was a record indicating an admission to hospital during the period of discontinuation – wondered how reliable the record, the best way is to use HES data?	Unfortunately, we did not have sufficient funds for HES data. Linked HES data is only available for a proportion of CPRD patients so whilst it may have provided some additional information on hospital admission it is unlikely to have improved precision significantly.	N/A
8. P.7 Can the authors suggest examples of codes indicating supporting the presence of the disease?	Yes, we would count codes such as attendance at specialist clinics or codes for disease follow up as supporting the presence of the disease.	Page 8: Added to section “Covariate classification”. Page 9: More specific examples are also included under specific covariates e.g. congestive heart failure.
9. Concomitant medications – do they have to have overlapped duration with OACs? Which medications will be investigated?	Ideally concomitant medications would overlap with OAC, however, due to difficulties knowing exactly when medications have been stopped we decided just to include any medication prescribed in the 3 months prior to the OAC. We assume that it may have affected the decision to prescribe the OAC whether or not it was still ongoing at the exact time the OAC was started which may be untrue in some cases but we have now added this as a limitation of the study.	Page 8: Medications to be investigated have been added to “Covariate classification” section. Page 14: Detail added to discussion section.
10. P.8 Hypertension – apart from Read codes, how about blood pressure records?	We have included both Read codes for hypertension and antihypertensive medication so feel that this is sufficient to capture the majority of patients with hypertension. Test records are not well populated so are unlikely to indicate many	N/A

	additional patients with hypertension that is not recorded elsewhere.	
11. P.12 Missing data will be assessed for randomness using a validated method – can the authors provide a citation?	Thank you for suggesting that we be more explicit about the methods used to handle missing data.	Page 14: Additional detail and references added to “Missing data” section.

VERSION 2 – REVIEW

REVIEWER	Madeleine Durand Centre Hospitalier de l'Université de Montréal Université de Montréal Canada
REVIEW RETURNED	22-Nov-2019

GENERAL COMMENTS	I thank you very much for your very thorough answers to my questions. I found the manuscript much clearer in its actual version. As a side note, I also completely agree with your choice of not using office BP measurements as 1)too many missing values and, 2)they are not a good way to diagnose HPB, mostly you would catch white coat hypertension. Good luck in conducting this ambitious and important study. I am looking forward to read your study, your results will be very interesting!
---